# Synthesis of tertiary alkylphosphonate oligonucleotides through light-driven radical-polar crossover reactions

Kenji Ota[1], Kazunori Nagao[1] ✉, Dai Hata [ID][2] ✉, Haruki Sugiyama[3,4,5], Yasutomo Segawa [ID][3,5], Ryosuke Tokunoh[2], Tomohiro Seki[2], Naoya Miyamoto[2], Yusuke Sasaki[2] & Hirohisa Ohmiya [ID][1,6] ✉

Chemical modification of nucleotides can improve the metabolic stability and target specificity of oligonucleotide therapeutics, and alkylphosphonates have been employed as charge-neutral replacements for naturally-occurring phosphodiester backbones in these compounds. However, at present, the alkyl moieties that can be attached to phosphorus atoms in these compounds are limited to methyl groups or primary/secondary alkyls, and such alkylphosphonate moieties can degrade during oligonucleotide synthesis. The present work demonstrates the *tertiary* alkylation of the phosphorus atoms of phosphites bearing two 2'-deoxynucleosides. This process utilizes a carbocation generated via a light-driven radical-polar crossover mechanism. This protocol provides *tertiary* alkylphosphonate structures that are difficult to synthesize using existing methods. The conversion of these species to oligonucleotides having charge-neutral alkylphosphonate linkages through a phosphoramidite-based approach was also confirmed in this study.

Recently, oligonucleotides have attracted attention with regard to pharmaceutical applications. These compounds exhibit high specificity for certain molecules and can allow the delivery of small-molecule drugs and antibodies to targets that are otherwise challenging to access[1,2]. Although naturally-occurring oligonucleotides are susceptible to degradation by endogenous nucleases, the chemical modification of oligonucleotides can be used to mitigate this instability and provide pharmaceuticals with high specificity for mRNA and miRNA[3,4]. In fact, chemically-modified oligonucleotides have been approved and marketed over the last two decades[5]. Oligonucleotides are composed of contiguous nucleotide subunits each having a sugar-based scaffold, and so the phosphodiester backbones and nucleobases of these compounds have been examined as potential sites of chemical modification[6,7]. However, while a variety of sugar-modified nucleic acids have been developed, the range of backbone modifications remains limited.

Chemical modifications of the phosphodiester backbone directly affect the metabolic stability of these compounds and also alter their pharmacokinetic profiles[8]. The formation of phosphorothioate (PS) groups is an example of a typical phosphodiester backbone modification and is found in the majority of commercially available oligonucleotide therapeutics[9]. This modification involves the replacement of the non-bridging oxygen atom in a phosphodiester bond with a sulfur atom as a means of increasing metabolic stability and lipophilicity. However, PS-modified oligonucleotides can exhibit toxicity as a consequence of nonspecific interactions between the negatively charged sulfur atoms and the positively charged proteins that are ubiquitous in living systems[10]. Charge-neutral backbone modifications are expected to maintain a high degree of metabolic stability while avoiding this toxicity. Present approaches to obtaining charge-neutral backbones comprise triester-type P–O linkages, several P–N linkages (such as phosphorodiamidate backbones in phosphorodiamidate

[1]Institute for Chemical Research, Kyoto University, Uji, Kyoto, Japan. [2]Research, Takeda Pharmaceutical Company Limited, Fujisawa, Kanagawa, Japan. [3]Institute for Molecular Science Myodaiji, Okazaki, Japan. [4]Comprehensive Research Organization for Science and Society Neutron Industrial Application Promotion Center, Tokai, Ibaraki, Japan. [5]Graduate Institute for Advanced Studies, SOKENDAI, Myodaiji, Okazaki, Japan. [6]JST, PRESTO, 4-1-8 Honcho, Kawaguchi, Saitama, Japan. ✉e-mail: nagao.kazunori.4j@kyoto-u.ac.jp; dai.hata@takeda.com; ohmiya@scl.kyoto-u.ac.jp

morpholino oligomers (PMO) and phosphoranylguanidine backbones) and P–C linkages, such as in the case of methylphosphonate (MP) and methoxypropylphosphonate (MOP) moieties (Fig. 1A)[5,6,11]. It should be noted that P–C backbones can be associated with challenges related to chemical stability, although high levels of nuclease resistance following the bonding of alkyl groups not found in natural products to phosphorus atoms has been demonstrated[12,13]. On this basis, we anticipated that sterically-hindered tertiary alkylphosphonates could serve as robust charge-neutral backbones ensuring high chemical stability. The conventional synthesis of oligomers containing P–C backbones involves the preparation of phosphoramidites in which a P–C bond is already introduced. However, to the best of our knowledge, there are presently no techniques for introducing bulky, multi-substituted carbon groups onto a phosphorus atom and then bonding this atom to a phosphoramidite. Moreover, steric hindrance effects might be expected to prohibit the use of tertiary alkylphosphonates in this coupling step. Because alternative approaches to oligonucleotide synthesis have not yet been unexplored, there is a need to develop a

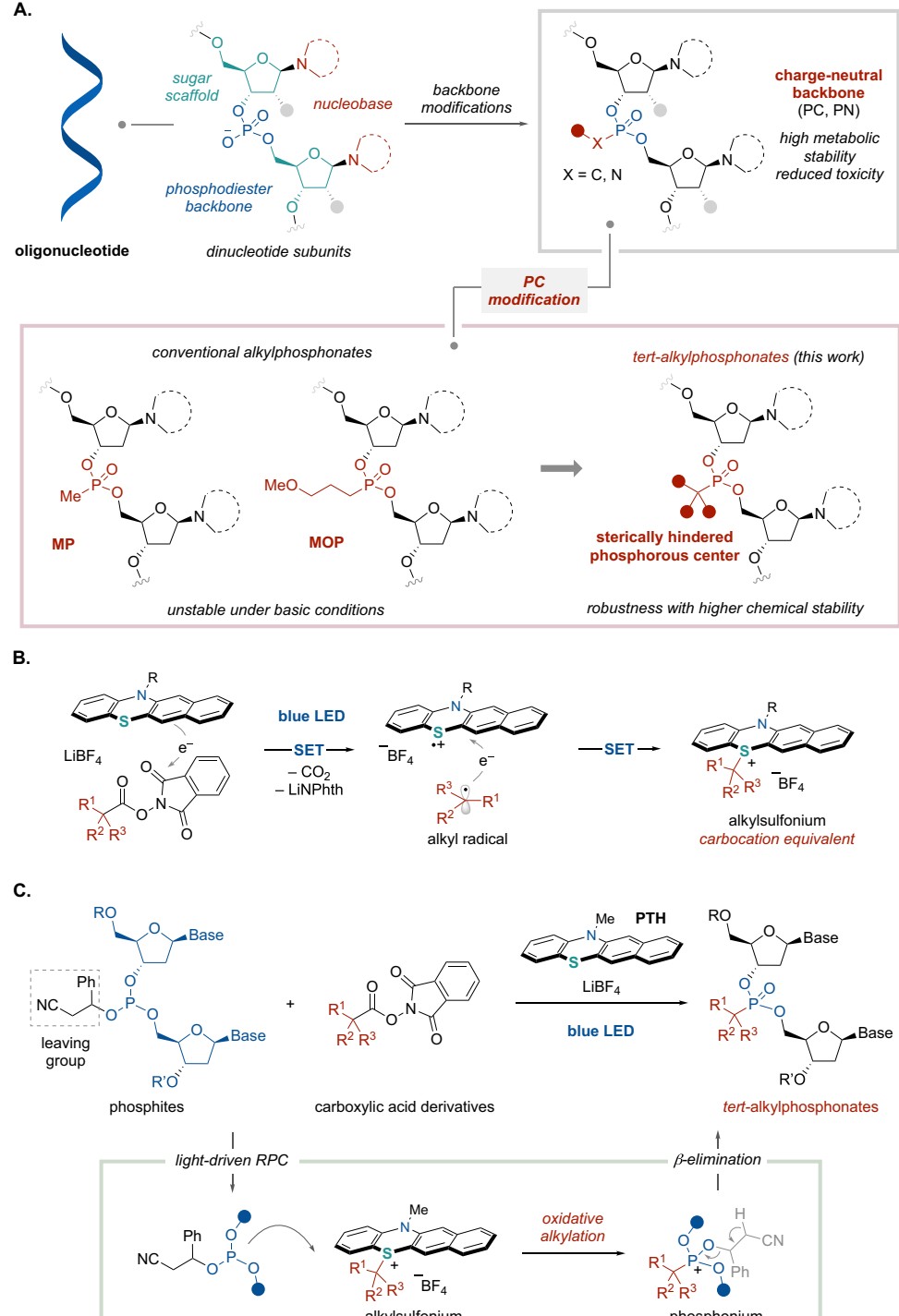

**Fig. 1 | Chemical modification of oligonucleotides. A** Applications of P–C backbones in oligonucleotide pharmaceuticals. MP methylphosphonate. MOP Methoxypropylphosphonate. **B** Carbocation generation via a light-driven radical-polar crossover mechanism. SET, single electron transfer. **C** Synthesis of oligonucleotides bearing *tertiary* alkylphosphonate backbones (this work).

fundamentally different approach that is compatible with a high degree of steric hindrance and with a wide range of functional groups.

Carbocations are highly reactive and hence can be employed to attach bulky alkyl substituents to heteroatom centers[14]. However, present-day methods require the use of strong acids to generate the carbocation species, which limits the range of functional groups that can be accepted[15]. Newer electrochemical and photochemical approaches have enabled the generation of carbocations under milder conditions without the use of strong acids as a means of forging C(sp³)–heteroatom bonds with various heteroatom nucleophiles[16–20]. Our own group previously developed a light-driven radical-polar crossover (RPC) protocol that combines visible light-mediated photoredox catalysis with an RPC mechanism and requires only mild conditions (Fig. 1B)[21–27]. In this process, a photo-excited benzo[b]phenothiazine donates an electron to a redox-active ester derived from a carboxylic acid in response to visible light irradiation to form a benzo[b]phenothiazine radical cation along with an alkyl radical, with the liberation of carbon dioxide. Subsequently, the alkyl radical is oxidized by the radical cation with the simultaneous combination of these species to form an alkylsulfonium compound. This alkylsulfonium species can subsequently react with a nucleophile, serving as a carbocation analogue. This method permits various tertiary and secondary alkyl substituents to be attached to heteroatom nucleophiles without the use of strong acids, such that a number of different highly functionalized molecules can be obtained. We envisioned that the carbocation equivalent (that is, the alkylsulfonium) generated by the light-driven RPC protocol could be applied to the tertiary alkylation of the phosphorus atoms of oligonucleotides. To assess the viability of this process, the present work examined the Michaelis-Arbuzov-type alkylation reactions of phosphites bearing two deoxynucleosides and a suitable leaving group with aliphatic carboxylic acid-derived redox-

active esters via a visible-light-driven RPC mechanism (Fig. 1C). This reaction is believed to proceed via the nucleophilic attack of the phosphorus atom of the phosphite on the carbocation equivalent, followed by a β-elimination associated with the loss of the phosphonium species[28]. Recently, several alkylphosphonates have been prepared by electrochemical or photochemical approaches involving reactions with radicals or carbocation species and phosphites[17,29–31]. However, the phosphites that can be applied to such protocols are often limited to simple substrates such as triethyl phosphite and so the application of this concept to the chemical modification of phosphorus atoms in complex oligonucleotides has not yet been demonstrated.

## Results and discussion
### Development of the reaction

After an extensive evaluation of reaction conditions based on our previous work[21,24], the use of stoichiometric amounts of N-phenylbenzophenothiazine (**PTH1**) and LiBF₄ was found to promote decarboxylative tertiary alkylation using a phosphite (**1a-1**) bearing two thymidine moieties and a 2-cyano-1-phenylethyl group as a leaving group together with a 2-phenylisobutyric acid-derived redox active ester (**2a**). This reaction was promoted by exposure to a blue light-emitting diode (LED) for 18 h and was performed in a mixture of acetonitrile (MeCN) and dichloromethane (DCM) to provide tertiary alkylphosphonate **3aa** in a 71% yield (Table 1, entry 1). Since phosphite **1a-1** was used as a mixture of diastereomers and each P-stereoisomer of the substrate was converted to the corresponding P-stereoisomer of the product, **3aa** was obtained as a mixture of diastereomers. After separation using a silica gel column chromatography, the structure of one of the diastereomers was determined by X-ray analysis. This analysis showed that the phosphorus center had an $R_P$ conformation.

## Table 1 | Screening of leaving groups[a]

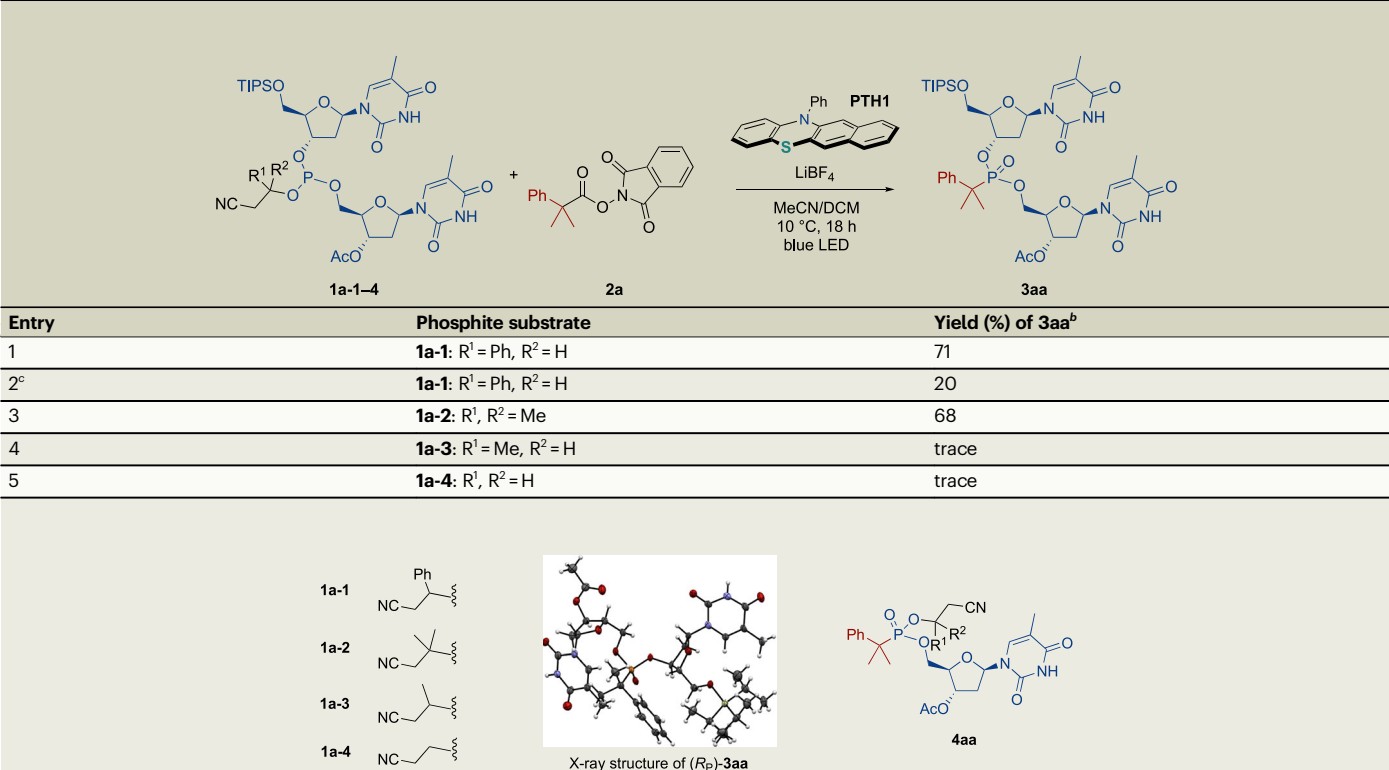

| Entry | Phosphite substrate | Yield (%) of 3aa[b] |
|---|---|---|
| 1 | **1a-1**: R¹ = Ph, R² = H | 71 |
| 2[c] | **1a-1**: R¹ = Ph, R² = H | 20 |
| 3 | **1a-2**: R¹, R² = Me | 68 |
| 4 | **1a-3**: R¹ = Me, R² = H | trace |
| 5 | **1a-4**: R¹, R² = H | trace |

X-ray structure of ($R_P$)-**3aa**

[a] Each reaction was carried out with **1** (0.05 mmol), **2a** (0.15 mmol), **PTH1** (0.05 mmol) and LiBF₄ (0.05 mmol) in MeCN (0.3 mL) and DCM (0.2 mL) at 10 °C under blue LED irradiation for 18 h[b].H NMR yields[c].**PTH1** (10 mol%) was used.

Interestingly, the use of a catalytic amount of **PTH1** decreased the yield (entry 2), possibly due to the low nucleophilicity of the phosphite.

The use of a 2-cyano-1,1-dimethylethyl leaving group (**1a-2**)[32] gave similar results to those of the earlier trials (Table 1, entry 3). In contrast, a 2-cyano-1-methylethyl (**1a-3**) and a 2-cyanoethyl moiety (**1a-4**) afforded **4aa** instead of the desired product **3aa** (entries 4 and 5). This result indicated that deprotonation of the phosphonium intermediate generated from **1a** with the simultaneous formation of a carbocation occurred at the 2′-position of the ribose molecule rather than at the α-cyano position on the leaving group[33]. For this reason, a phenyl (**1a-1**) or 1,1-dimethyl (**1a-2**) substituent was expected to preferentially induce deprotonation at the α-cyano position.

The effects of reaction components were subsequently evaluated, using **1a-1** as the phosphite substrate (Table 2). Initially, various substituents were appended to the nitrogen atom on the benzo[*b*]phenothiazine. The use of N-methyl benzo[*b*]phenothiazine (**PTH2**) slightly increased the yield while N−H benzo[*b*]phenothiazine (**PTH3**) significantly decreased the yield (Table 2, entries 1–3). In contrast to **PTH1**, which was synthesized via a Buchwald-Hartwig amination, **PTH2** could be prepared without requiring a transition metal-catalyzed amination and so this reagent is also advantageous in that respect[34]. The benzo[*b*]phenothiazine core was found to be essential to this reaction (entry 4).

The effects of various additives and solvents were investigated while employing **PTH2** as the redox mediator. In the absence of $LiBF_4$ or when this reagent was replaced by $LiPF_6$ or $NaBF_4$, the target

product was not obtained (Table 2, entries 5–7). These results demonstrate that the presence of both Li cations and $BF_4$ anions was important in this reaction. The reaction proceeded in either MeCN or DCM as the sole solvent although lower yields were obtained (entries 8 and 9). The use of a more polar solvent such as DMF did not yield the desired product at all while the minimally polar solvent THF resulted in lower yields (entries 10 and 11). When the reaction temperature was lowered or raised from 10 °C, the yield was also diminished (entries 12–14).

## Substrate scope

With the optimal reaction conditions in hand, the ranges of phosphites and redox-active esters that could be employed were investigated (Fig. 2). Various phosphites were initially examined together with a 2-phenylisobutyric acid derivative (**2a**). Phosphites bearing two different 2′-deoxynucleoside scaffolds such as deoxycytidine (**3ba**–**3da**), deoxyadenosine (**3ea**) and deoxyguanosine (**3fa**) were found to participate in this P-alkylation protocol to afford the corresponding alkylphosphonates. The relatively low yields obtained with **3ea** and **3fa** could possibly have resulted from the low oxidation potentials of the purine bases[35]. The reaction proceeded smoothly even with a *tertiary* butyldimethylsilyl (TBS) group as the 3′-OH protecting group (**3ca**).

The scope of alkyl substituents was subsequently examined using **1a-1** as the nucleophile. It was found to be possible to apply aliphatic carboxylic acid derivatives with halogen substituents to the benzene ring as sites for further molecular transformations (**3ab**)[36]. Redox-

## Table 2 | Screening of reaction conditions[a]

| Entry | Change from standard conditions | Yield (%) of 3aa[b] |
|---|---|---|
| 1 | None | 71 |
| 2 | **PTH2** instead of **PTH1** | 74 (70) |
| 3 | **PTH3** instead of **PTH1** | 37 |
| 4 | **PTH4** instead of **PTH1** | 0 |
| 5[c] | Without $LiBF_4$ | 0 |
| 6[c] | $LiPF_6$ instead of $LiBF_4$ | trace |
| 7[c] | $NaBF_4$ instead of $LiBF_4$ | 0 |
| 8[c] | MeCN instead of MeCN/DCM | 45 |
| 9[c] | DCM instead of MeCN/DCM | 53 |
| 10[c] | DMF instead of MeCN/DCM | 0 |
| 11[c] | THF instead of MeCN/DCM | 39 |
| 12[c] | 0 °C instead of 10 °C | 31 |
| 13[c] | 20 °C instead of 10 °C | 69 |
| 14[c] | 40 °C instead of 10 °C | 65 |

[a] Each reaction was carried out with **1a-1** (0.05 mmol), **2a** (0.15 mmol), **PTH1** (0.05 mmol) and $LiBF_4$ (0.05 mmol) in MeCN (0.3 mL) and DCM (0.2 mL) at 10 °C under blue LED irradiation for 18 h[b1].H NMR yields. The number in parentheses is the isolated yield[c].**PTH2** (1.0 equiv.) was used.

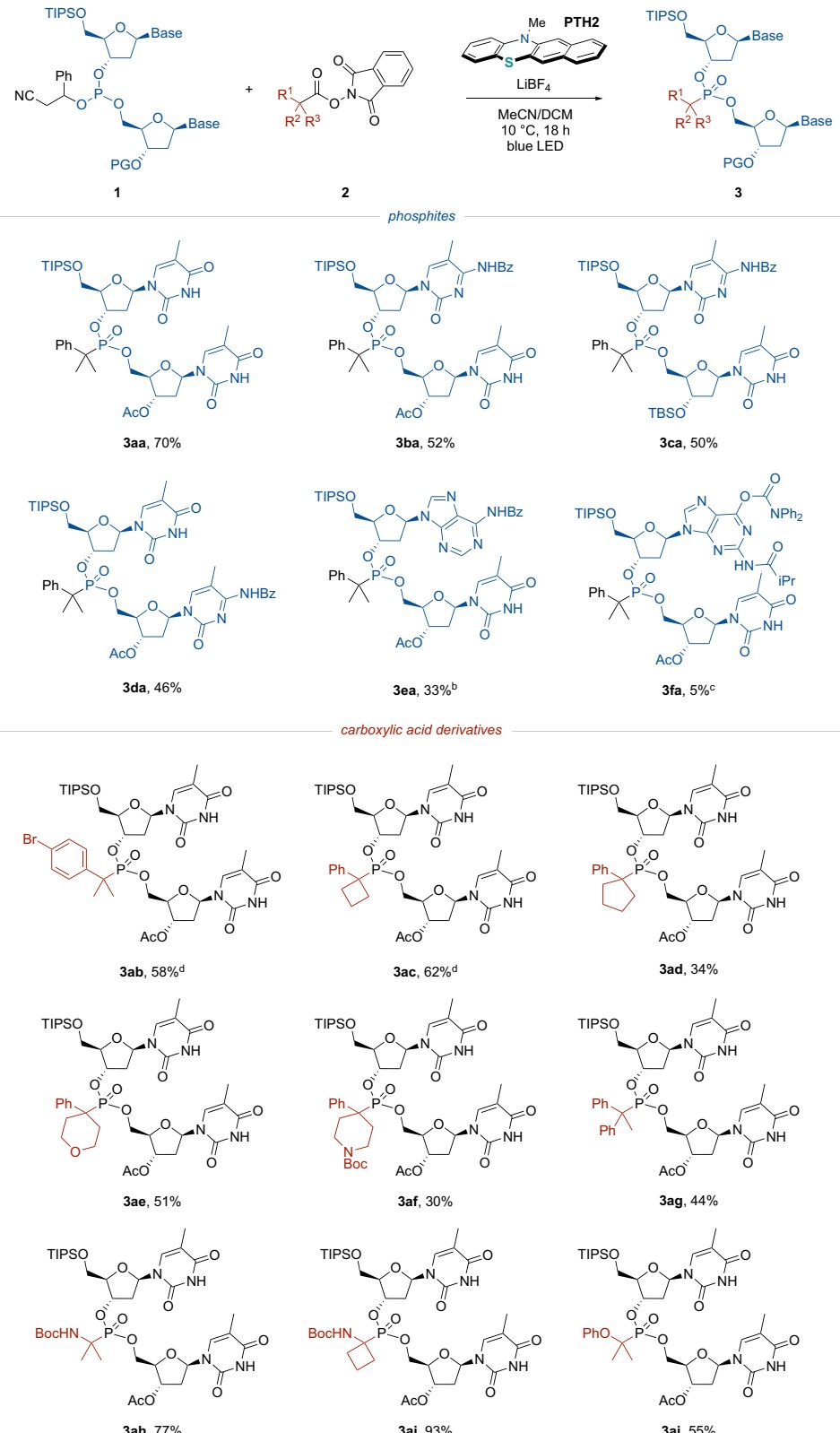

**Fig. 2 | Substrate scope.** Each reaction was carried out with **1** (0.05 mmol), **2** (0.15 mmol), **PTH2** (0.05 mmol) and LiBF$_4$ (0.05 mmol) in MeCN (0.3 mL) and DCM (0.2 mL) at 10 °C under blue LED irradiation for 18 h[a] .**PTH2** (1.5 equiv.), LiBF$_4$ (1.5 equiv.) and RAE (5.0 equiv.) were used [b] **PTH2** (1.5 equiv.), LiBF$_4$ (2.0 equiv.) and RAE (5.0 equiv.) were used[c]. Each reaction was carried out at 20 °C.

active esters with various cyclic carbon skeletons were also employed and gave the desired alkylphosphonates in good yields (**3ac** and **3ad**). In addition, heterocyclic structures such as tetrahydropyran and piperidine could be attached to the phosphorus atom (**3ad** and **3af**) and a relatively bulky diphenylmethyl group was also tolerated (**3ag**). Furthermore, the reaction was determined to proceed without issue both at the benzylic position and at the α-positions of heteroatoms such as nitrogen and oxygen atoms (**3ah–3aj**).

## Control experiments

Additional insights into this light-driven RPC protocol were obtained via a series of experiments. A stoichiometric reaction between **PTH2** and **2a** in the presence of LiBF$_4$ was initially conducted to observe the key alkylsulfonium intermediate (Fig. 3A). Following irradiation with a blue LED for 4 h, the formation of an alkylsulfonium species was confirmed by analysis of the reaction solution using electro spray ionization–high resolution mass spectrometry. However, attempts at additional characterization by nuclear magnetic resonance spectroscopy or isolation of the alkylsulfonium were unsuccessful. Nevertheless, in the case that the reaction was performed for the same duration as used in the earlier P-alkylation reactions and with stoichiometric quantities of reagents, the majority of the original **2a** was recovered. This observation prompted an investigation of the effect of the various reaction components on the conversion of **2a**. In the absence of the phosphite nucleophile **1a-1**, the conversion of the redox-active ester **2a** was found to be only 11% after 4 h. Conversely, after **1a-1** was added, the majority of the **2a** was converted after 4 h with a 96% yield. This result suggests that **1a-1** affected the conversion of the redox-active esters. Previous reports have stated that Brønsted acids can facilitate single electron transfer from a photoredox catalyst to a redox active ester, resulting in the efficient formation of an alkyl radical[23,27,37]. Considering that the N−H group of the thymine moiety might act as a Brønsted acid, the reaction was carried out in the presence of thymine. As expected, the conversion of **2a** with an increased 28% yield was observed.

The synthetic utility of this protocol was further assessed by examining whether previously reported protocols for carbocation generation could be applied to the tertiary alkylation of **1a-1** (Fig. 3B). The 4CzIPN-based photoredox catalysis using aliphatic redox-active esters as reported by Aggarwal and co-workers was initially employed in conjunction with **1a-1**. However, the addition of trifluoroacetic acid was found to decompose the **1a-1** such that none of the target product **3aa** was obtained. Other standard methods utilizing a Lewis acid catalyst and an alcohol as the source of the alkyl group were also investigated. These trials employed trimethylsilyl trifluoromethanesulfonate or zinc iodide (ZnI$_2$) together with 2-phenyl-2-propanol (**5a**) and **1a-1** while also heating the reaction solution[38,39]. Unfortunately, none of the intended product (the alkyl phosphonate **3aa**) was produced but rather a complex mixture of various compounds was generated. These results indicate that generation of the carbocation in the present protocol does not requires the addition of strong acids as mediators, which can lead to decomposition of the phosphite, thus permitting chemoselective alkylation.

## Automated solid-phase synthesis of oligonucleotides

The synthesis of oligonucleotides having backbone structures comprising advantageous bulky alkylphosphonate dimers was also demonstrated (Fig. 4). Initial attempts involved the synthesis of 5′-O-4,4′-dimethoxytrityl (DMTr)-phosphoramidite (**3aa−3**) from **3aa** (Fig. 4A), during which the isolated diastereomers ($R_P$)−**3aa** and ($S_P$)-

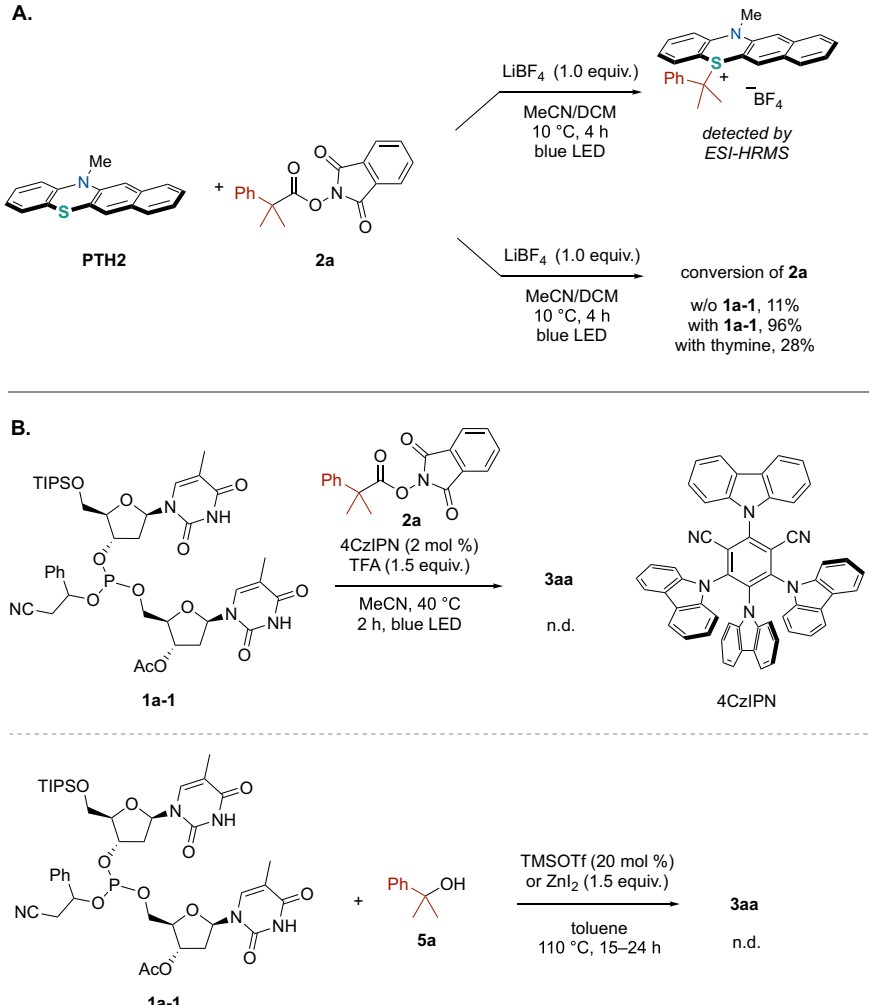

**Fig. 3 | Control experiments. A** Stoichiometric reaction between **PTH2** and **2a**. **B** Comparison with other protocols involving carbocation generation.

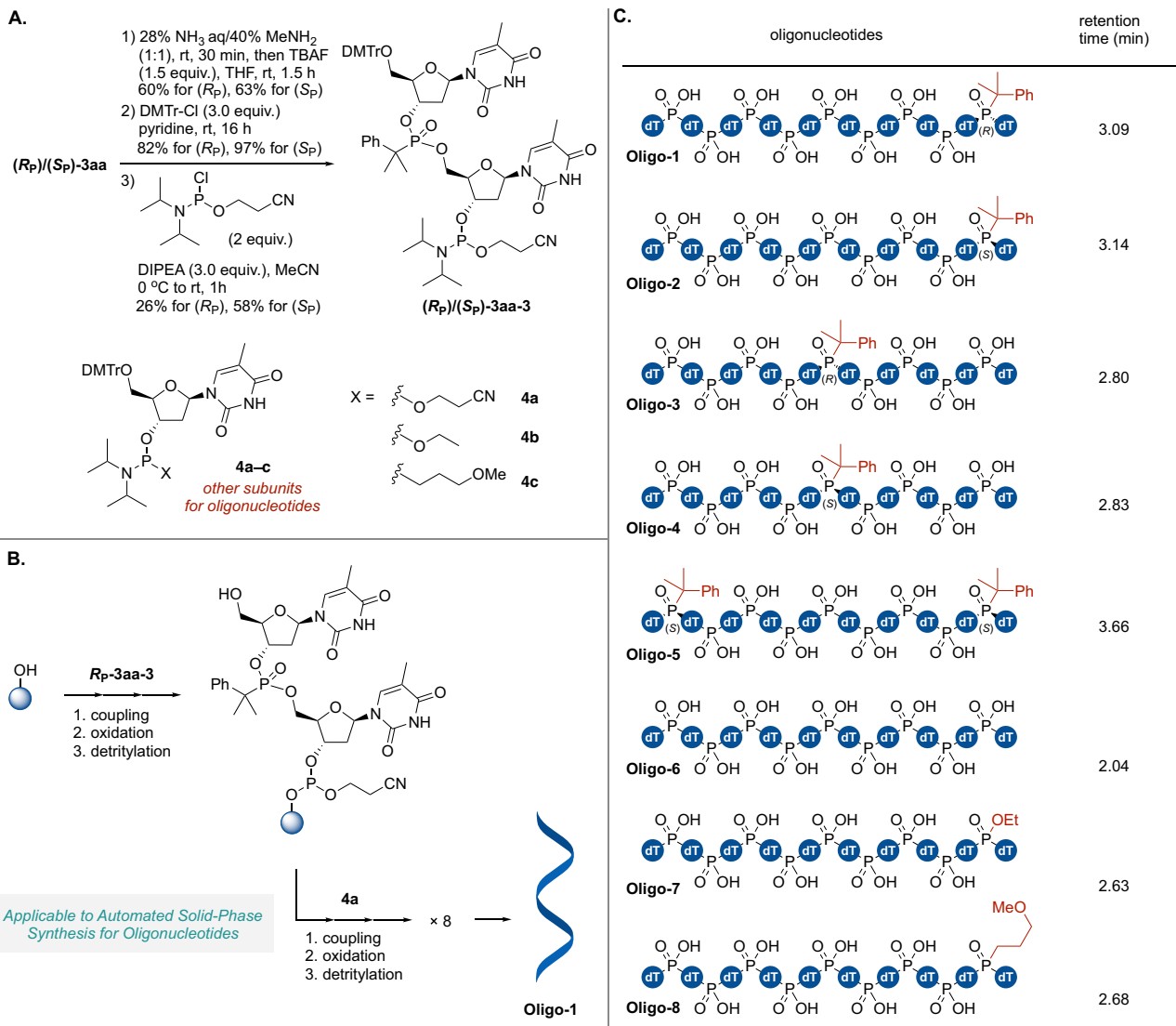

**Fig. 4 | Application of the present method to the synthesis of oligonucleotides.** **A** Derivatization of $(R_P)/(S_P)$-**3aa** to give the phosphoramidites $(R_P)/(S_P)$-**3aa-3**. **B** Solid-supported automated synthesis of oligonucleotides. The synthesis of **Oligo-1** was exemplified. **C** Retention times obtained from liquid chromatography analysis of synthesized oligonucleotides.

**3aa** were converted to the corresponding phosphoramidites. The 5′-O-acetyl and 3′-O-TIPS protecting groups of **3aa** were removed in series without purification to give the corresponding diols $(R_P)$-**3aa-1** and $(S_P)$-**3aa-1**. These diols were then reacted with DMTr-Cl to protect the 5′-OH group and obtain the free 3′-OH products $(R_P)$–**3aa-2** and $(S_P)$–**3aa-2**. Following this, a reaction with 2-cyanoethyl *N,N*-diisopropyl-chlorophosphoramidite gave phosphoramidites with $R_P$ or $S_P$ phosphorous centers for oligonucleotide synthesis [$(R_P)$-**3aa-3** and $(S_P)$-**3aa-3**]. The bulky alkylphosphonate backbone was found to remain stable in the presence of nucleophiles such as ammonia, methylamine and tetrabutylammonium fluoride (TBAF) and to be unaffected by the basic pH values imparted to the reaction solution by these amines.

The solid-phase synthesis of oligonucleotides having bulky alkyl-phosphonate backbones was achieved by employing the widely-used phosphoroamidite method together with an automated oligo-synthesizer (Fig. 4B)[40–44]. In this process, reactive phosphorus (III) compounds were incorporated into the growing oligonucleotide chain through a cycle of coupling, oxidation, and deprotection of the DMTr group steps. The synthesis cycle was repeated until the desired chain length was reached and the target oligonucleotide was then obtained through cleavage and deprotection. Using the phosphoroamidite

**3aa-3** and a 5-O′-DMTr-thymidine (dT)-derived phosphoroamidite bearing a 2-cyanoethoxy group (**4a**), several oligonucleotides (**Oligo-1–6**) were synthesized using the automated oligo-synthesizer with a solid support (Fig. 4C). For comparison with other neutral backbones, ethyl triester-modified **Oligo-7** and MOP-modified **Oligo-8** were prepared from **4b** and **4c**, respectively. Using the optimized conditions for the synthesizer, the oligonucleotides (**Oligo-1–5**) having bulky alkyl-phosphonate backbones were synthesized in the comparable yields with **Oligo-6** comprising of only naturally-occurring phosphodiester backbones (Supplementary Fig. 4). Analysis by liquid chromatography prior to purification provided similar results in each case, confirming that there was no degradation as a consequence of the backbone structure during the solid phase synthesis or the cleavage and deprotection processes. These results confirm that the backbone structure produced in the present work was robust and that the ter-tiary alkylphosphonate structure was evidently unreactive in the pre-sence of various electrophiles and acidic conditions during the solid phase synthesis. As demonstrated by the synthesis of the phosphor-oamidite dimers, the oligomers also exhibited resistance to the basic conditions applied during the cleavage and deprotection processes. In addition, the product could be readily purified using a standard

cartridge-type kit. Analyses by reverse phase liquid chromatography indicated that the retention time for the $R_P$ form was longer than that for the $S_P$ form, suggesting that the former was more globally hydrophobic. The retention times of **Oligo-1** and **Oligo-2** were also greater than those for **Oligo-7** and **8** and the difference between the retention times of the $S_P$ and $R_P$ configurations of these compounds was expanded (Supplementary Fig. 4). The effect of chirality on global hydrophilicity/hydrophobicity was therefore increased as the molecules became larger. This effect could also modify the protein binding profiles of these compounds and lead to eutomer/distomer differences in the case that they are utilized as pharmaceuticals. The introduction of the alkyl group at the 3′ end of the oligomer had a greater effect on retention time than introduction at the center (**Oligo-1**–**4**). The physicochemical and biological effects resulting from the formation of these unique tertiary alkylphosphonate backbones are currently under investigation by our group.

In summary, this work demonstrated a synthetic protocol enabling the tertiary alkylation of phosphites bearing two 2′-deoxyynucleosides, using N-methyl benzo[*b*]phenothiazine and tertiary aliphatic carboxylic acid-derived redox-active esters in conjunction with irradiation by a blue LED. This process was found to allow the formation of bulky tertiary alkylphosphonate structures that are difficult to synthesize by conventional methods. The reaction evidently proceeds via a light-driven RPC mechanism involving carbocation species and phosphites. The resulting tertiary alkylphosphonate structures exhibit significant stability in response to various environments such that this process can be used for oligomer synthesis.

## Methods

### The reaction in Table 2, entry 2 is representative

In a glovebox, to an oven-dried vial with a stirring bar was added **PTH2** (13.2 mg, 0.05 mmol), phosphite **1a-1** (42.9 mg, 0.05 mmol), 2-phenylisobutyric acid-derived redox active ester **2a** (46.4 mg, 0.15 mmol) and LiBF$_4$ (4.7 mg, 0.05 mmol). Then, MeCN (300 μL) and DCM (200 μL) were added to the reaction mixture. After sealing the vial with parafilm, the reaction mixture was stirred and irradiated with a 34 W blue LED with UC reactor to keep the temperature at 10 °C (Supplementary Fig. 2). After 18 h, the solvents were removed under reduced pressure. The crude material was then purified by flash column chromatography on silica gel (80:20:0–50:45:5, hexane/EtOAc/MeOH) to give the alkylated product **3aa** (29.6 mg, 0.035 mmol, 70% isolated yield) as a white amorphous solid.

## Data availability

The authors declare that the data supporting the findings of this study are available within the paper or its Supplementary Information files and from the corresponding author upon request. The X-ray crystallographic coordinates for the structure reported in this study have been deposited at the Cambridge Crystallographic Data Centre (CCDC), under deposition number 2263561. This data can be obtained free of charge from The Cambridge Crystallographic Data Centre via www.ccdc.cam.ac.uk/data_request/cif.

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

## Acknowledgements

This work was supported by the Takeda Pharmaceutical Company, Ltd. and by JSPS KAKENHI grants (nos. JP21H04681, JP23H04912, and JP22KJ1938) and a JST PRESTO grant (no. JPMJPR19T2). A part of this work was conducted at the Institute for Molecular Science supported by ARIM (JPMXP1223MS5012).

## Author contributions

K.O., K.N., D.H., and H.O. conceived the project. K.O., D.H., K.N., R.T., T.S., N.M., Y. Sa., and H.O. performed the experiments and analyzed the data. H.S and Y. Se. performed X-ray crystallography analyses. K.O., K.N., D.H., and H.O. co-wrote the manuscript. All authors contributed to discussions.

## Competing interests

The authors declare no competing interests.
