## [Peer Review File · Nature Communications]

Synthesis of tertiary alkylphosphonate oligonucleotides through light-driven radical-polar crossover reactionsREVIEWER COMMENTS

Reviewer #1 (Remarks to the Author):

see attached review

Recommendation: Publish in Nature Communications after addressing the minor revisions

Comments: Nagao, Hata, Ohmiya and co-workers demonstrated an interesting synthetic route for the tertiary alkylation of phosphites substituted with two 2'-deoxynucleosides, using *N*-methyl benzo[*b*]phenothiazine and tertiary aliphatic carboxylic acid-derived redox-active esters generating bulky tertiary alkyl phosphonates, which was formerly a challenging process. They have also applied this method for oligomer synthesis through a cycle of coupling, oxidation, and deprotection sequence. The overall process is useful and, more importantly, it addresses the major limitation of Arbuzov reactions that were mostly limited to the primary and secondary alkyl radicals. The paper is excellent.

Minor revisions:

1. For a phosphite substrate variation **1a-1 to 1a-3**, it would be logical to include a methyl substituted example in place of a phenyl to see the pattern.
2. It is quite unusual that this reaction works very well at 10 °C, but not well at 40 °C. Where is the room temperature result? Is something decomposing at higher temperature? Low reactivity can be understood when going down to 0 °C.
3. Include the following very recent citations for primary and secondary alkyl radical phosphonylation where appropriate: (i) Phosphonylation of alkyl radicals: Junyue Yin, Xinru Lin, Linxiang, Chai, Cheng-Yu Wang, Lin Zhu, Chaozhong Li, *Chem* **2023**, 9 (7), 1945-1954. <https://doi.org/10.1016/j.chempr.2023.03.016> (ii) Convergent Deboronative and Decarboxylative Phosphonylation Enabled by the Phosphite Radical Trap "BecaP" Santosh. K. Pagire, Chao Shu, Dominik Reich, Adam Noble, Varinder K. Aggarwal, *J. Am. Chem. Soc.* **2023**, ASAP, <https://doi.org/10.1021/jacs.3c06524>

Supporting information:

The proper characterization of most of the compounds has been carried out. However, following things need to be addressed before publishing: the coupling constant (J_{C-P}) values of phosphite and phosphonates should be included in the ^{13}C NMR description – it might be difficult to report everything correctly as it is a diastereomeric mixture (more scans/higher MHz machine/more compound amount would make C^{13} NMRs look clear and thus "J" values can easily be found). Nice to see that it is reported for a few compounds: e.g. **(Rp)-3aa-1**. It can be seen that most of the ^{13}C NMR experiments are not properly done and some signals are hidden in the noise/baseline e.g. **1b, 3ae, 3af, (Rp)-3aa-3 ...etc.** Also, ^{31}P NMR should look very sharp and it appears for a few compounds it's not (e.g. **1c, 1d, 3af**, etc). – there is a lot of noise in the baseline. Authors should pay special attention in getting clean and sharp ^{31}P (as well as ^{13}C) spectra. For instance, acceptable ^{13}C and ^{31}P NMR spectra are found for the compounds **(Sp)-3aa-1** or **(Rp)-3aa-1**.

Reviewer #2 (Remarks to the Author):

The manuscript by Ohmiya and co-workers describes the development of a synthetic method to access dinucleotides with a tertiary alkylphosphonate internucleotide linkage. The method relies on a light-driven formation of alkylsulfonium derivative, which undergoes an in situ Arbuzov-type reaction with dinucleoside phosphite triester. In addition to the optimization of the reaction conditions and the exploration of its scope and limitations, the Authors demonstrate the feasibility of incorporation of the dinucleoside units with tertiary alkylphosphonate linkage into oligonucleotides. Thus, the paper needs to be evaluated at two levels: (1) the novelty of the developed synthetic method and (2) the importance and potential implications of the new modification of oligonucleotides.

From the chemistry viewpoint, the Authors employ their resourceful in-house approach of generating tertiary carbocation-like sulfonium species via the light induced activation of redox-active esters. It has been extensively explored earlier and shown to efficiently deliver tertiary alkyl groups to a variety of nucleophiles (not phosphorus-based, though). From this perspective, the presented results constitute only an incremental advancement and moderate originality, as basically yet another nucleophile family has been tested. Nevertheless, the method provides access to tertiary alkylphosphonates, which cannot be easily obtained otherwise. The only general method existing up to now toward this class of products is phospho-Michael addition, which is limited to compounds with electron-poor tertiary alkyl moiety, while the reported reaction allows for the incorporation of electron-rich groups. The Authors made a choice to optimize the reaction conditions directly for a dinucleotide phosphite substrate. This challenging starting material necessitated the use of a stoichiometric amount of benzo[b]phenothiazine reagent. Also, the obtained yields are mostly moderate. I do not fully agree with this approach, because the method could be made more general, and perhaps even catalytic for simpler phosphite esters (the corresponding products would also be of interest). In my opinion, this would benefit the chemical community more, rather than focusing on the nucleotide derivatives alone. Finally, a bit aside, I would like to ask if the Authors have tried or considered employing H-phosphonate diesters instead of phosphite triesters as the nucleophiles. The former have several advantages, such as straightforward synthesis (also in the case of dinucleotides) and rendering the auxiliary beta-cyanoalkyl substituent unnecessary.

Coming to the second aspect, the development of oligonucleotide therapeutics requires utilizing modified nucleotide units to attain desired pharmacological characteristics. This is of particular importance for the antisense and aptamer technologies, wherein, unlike in the case of mRNA vaccines, the enzymatic stability of the oligonucleotide needs to be considerably augmented. In this context, expansion of the toolbox of available modifications, especially these at the internucleotide phosphate linkage, is a viable direction of research. The tertiary alkylphosphonate moiety introduced in the work seems to bring about many favorable properties, including good hydrolytic stability and charge-neutrality. Very importantly from the practical perspective, the modified dinucleotide units can be efficiently incorporated into oligonucleotides via the standard phosphoramidite solid-phase synthesis (although a prior swap of protecting groups is necessary). Also, the high hydrolytic stability (and that toward F-) of the modification is nicely demonstrated. A clear highlight is the possibility to separate the P-diastereomers and use them individually for the modification of specific positions of the oligonucleotide. However, the actual usefulness of tert-Alk-P-modified oligos for biochemical and pharmaceutical applications is yet to be evaluated. One obvious foreseeable problem is that with this technology only some of the internucleotide linkages can be modified, while at least half of them will remain the regular phosphodiester bonds. In general, there has been a multitude of nucleotide modifications proposed and introduced, but only a handful made it to the therapeutic level.

In conclusion, the work by Ohmiya and co-workers is a valuable and interesting contribution. I appreciate employing cutting edge chemical methodology to access potentially biochemically-relevant molecules. However, in my opinion the presented chemistry is not particularly original and has low generality. The target compounds have been successfully prepared and incorporated into oligonucleotides, but at the moment it is impossible to tell the usefulness and significance of the novel

modification. Therefore, in my opinion the manuscript does not meet the criteria required for a publication in Nature Communications.

Irrespective of the future of the manuscript, I recommend following changes and improvements:

- 1) Abstract, line 7: "oligomerization" is not specific enough and should be replaced with "oligonucleotide synthesis". Also, these are not "methyl groups or primary/secondary alkyls" that degrade, but rather the phosphonate moieties.
- 2) Page 2, line 8: One cannot say that oligonucleotides are composed of "dinucleotide subunits", as there is nothing distinct about 2-nucleotide units (why not "trinucleotide subunits"?). Change to just "nucleotide" subunits.
- 3) Page 5, after the information about the diastereomers: The Authors may add a comment on the probable stereochemical course of the reaction. For instance, that probably each P-stereoisomer of the substrate is converted into the corresponding P-stereoisomer of the products, and as a mixture was used for the reaction, a mixture is obtained.
- 4) Table 1: It would improve the readability if a column listing the R1 and R2 substituents was added to the table
- 5) Figure 2: Please add the isolated yield of 3aa
- 6) Page 10: I would change the heading from "Mechanistic studies" to "Control experiments". True mechanistic studies would be more elaborate, like proper kinetic measurements, etc.

Reviewer #3 (Remarks to the Author):

In this manuscript, Ohmiya and co-workers describe a novel method of synthesizing P-tertiary alkyl-modified dinucleotides, realized through the nucleophilic attack of specially designed dinucleotide-bearing phosphite reagent to a stabilized tertiary carbocation generated under photocatalytic conditions. In this work, redox active ester was chosen as the tertiary radical precursor, and benzophenothiazine was used as the photocatalyst while acting as the key reagent to stabilize the tertiary cation through reversible formation of sulfonium salt. The stability provided by the sterically bulk tertiary alkyl-modified phosphonate was demonstrated by automated multi-step oligonucleotide synthesis. The method is novel and the practical usage for oligomer synthesis is obvious, I therefore recommend its publication in Nat. Commun., with following suggestions for minor revisions.

1. Have the authors tried used an external reversible-trapping reagent for the tertiary cation (instead of stoichiometric amount of photocatalyst)? Maybe screening of the trapping reagent could lead to better results.
2. Have the authors tried using a simpler phosphite, for example with a methoxy group, while adding an external nucleophile to abstract the methyl group from the phosphonium intermediate to enable the Arbuzov-type reactivity?
3. The substrate scope shows rather poor compatibility with trialkyl-substituted tertiary carboxylic acid. What are the results with these substrates? As I suggested above, maybe an extra trapping reagent could lead to the success with these substrates.
4. It will be nice if the authors can indicate any unsuccessful substrates, which will provide a clearer picture for the synthetic community.
5. Further potential mechanistic study could include distinguishing between outer-sphere single electron transfer mechanism and EDA mechanism. The benzophenothiazine is a pretty good donor and RAE is known to be good acceptor.
6. Some very closely related automated synthesis study should be cited, e.g. Nat. Commun. 2021, 12, 4396; Nat. Chem. 2021, 13, 451; ACS Cent. Sci. 2022, 8, 205.

Point-by-point Response to Reviewers' Comments

Reviewer 1

Nagao, Hata, Ohmiya and co-workers demonstrated an interesting synthetic route for the tertiary alkylation of phosphites substituted with two 2-deoxynucleosides, using Nmethyl benzo[b]phenothiazine and tertiary aliphatic carboxylic acid-derived redox-active esters generating bulky tertiary alkyl phosphonates, which was formerly a challenging process. They have also applied this method for oligomer synthesis through a cycle of coupling, oxidation, and deprotection sequence. The overall process is useful and, more importantly, it addresses the major limitation of Arbuzov reactions that were mostly limited to the primary and secondary alkyl radicals. The paper is excellent.

1. For a phosphite substrate variation 1a-1 to 1a-3, it would be logical to include a methyl substituted example in place of a phenyl to see the pattern.

Thank you for the valuable suggestion. According to the reviewer's suggestion, methyl substituted example (**1a-3**) has been added to the revised manuscript (Table1, Entry 4). Also, the corresponding comment has been added.

Table 1 | Screening of leaving groups^{a,d}

Entry ^c	Phosphite substrate ^c	Yield (%) of 3aa ^{b,c}
1 ^c	1a-1 : R ¹ = Ph, R ² = H ^c	71 ^c
2 ^c	1a-1 : R ¹ = Ph, R ² = H ^c	20 ^c
3 ^c	1a-2 : R ¹ , R ² = Me ^c	68 ^c
4 ^c	1a-3 : R ¹ = Me, R ² = H ^c	trace ^c
5 ^c	1a-4 : R ¹ , R ² = H ^c	trace ^c

The use of a 2-cyano-1,1-dimethylethyl leaving group (**1a-2**)^b gave similar results to those of the earlier trials (Table 1, entry 3). **In contrast, a 2-cyanoethyl moiety (1a-4), commonly employed as a protecting group for the phosphate during oligo- or dinucleotide synthesis, afforded 4aa instead of the desired product 3aa (entry 4).** In contrast, a 2-cyano-1-methylethyl (**1a-3**) and a 2-cyanoethyl moiety (**1a-4**) afforded **4aa** instead of the desired product **3aa** (entries 4 and 5). This result indicated that deprotonation of the phosphonium intermediate generated from **1a** with the simultaneous formation of a

2. It is quite unusual that this reaction works very well at 10 °C, but not well at 40 °C. Where is the room temperature result? Is something decomposing at higher temperature? Low reactivity can be understood when going down to 0 °C.

Thank you for the valuable suggestion. According to the suggestion, we tested the reaction at 20 °C and found that the yield was only slightly lower than at 10 °C (Table 2, Entry 13). As the reviewer pointed out, the reaction at higher temperatures results in lower yields due to increased decomposition of phosphite.

Table 2 | Screening of reaction conditions^a

Entry ^c	Change from standard conditions ^c	Yield (%) of 3aa ^b
1 ^c	None ^c	71 ^c
2 ^c	PTH2 instead of PTH1 ^c	74 (70) ^c
3 ^c	PTH3 instead of PTH1 ^c	37 ^c
4 ^c	PTH4 instead of PTH1 ^c	0 ^c
5 ^c	Without LiBF ₄ ^c	0 ^c
6 ^c	LiPF ₆ instead of LiBF ₄ ^c	trace ^c
7 ^c	NaBF ₄ instead of LiBF ₄ ^c	0 ^c
8 ^c	MeCN instead of MeCN/DCM ^c	45 ^c
9 ^c	DCM instead of MeCN/DCM ^c	53 ^c
10 ^c	DMF instead of MeCN/DCM ^c	0 ^c
11 ^c	THF instead of MeCN/DCM ^c	39 ^c
12 ^c	0 °C instead of 10 °C ^c	31 ^c
13 ^c	20 °C instead of 10 °C ^c	69 ^c
14 ^c	40 °C instead of 10 °C ^c	65 ^c

3. Include the following very recent citations for primary and secondary alkyl radical phosphonylation where appropriate: (i) Phosphonylation of alkyl radicals: Junyue Yin, Xinru Lin, Linxiang, Chai, Cheng-Yu Wang, Lin Zhu, Chaozhong Li, *Chem* 2023, 9 (7), 1945-1954. <https://doi.org/10.1016/j.chempr.2023.03.016> (ii) Convergent Deboronative and Decarboxylative Phosphonylation Enabled by the Phosphite Radical Trap “BecaP” Santosh. K. Pagire, Chao Shu, Dominik Reich, Adam Noble, Varinder K. Aggarwal, *J. Am. Chem. Soc.* 2023, ASAP, <https://doi.org/10.1021/jacs.3c06524>

According to the reviewer’s advice, the corresponding references have been added to the revised manuscript as references 30 and 31, respectively.

Supporting information: The proper characterization of most of the compounds has been carried out. However, following things need to be addressed before publishing: the coupling constant (J_{c-p}) values of phosphite and phosphonates should be included in the 13C NMR description – it might be difficult to report everything correctly as it is a diastereomeric mixture (more scans/higher MHz machine/more compound amount would make C13 NMRs look clear and thus “J” values can easily be found). Nice to see that it is reported for a few compounds: e.g. (Rp)-3aa-1. It can be seen that most of the 13C NMR experiments are not

properly done and some signals are hidden in the noise/baseline e.g. 1b, 3ae, 3af, (RP)-3aa-3 ...etc. Also, 31P NMR should look very sharp and it appears for a few compounds it's not (e.g. 1c, 1d, 3af, etc). – there is a lot of noise in the baseline. Authors should pay special attention in getting clean and sharp 31P (as well as 13C) spectra. For instance, acceptable 13C and 31P NMR spectra are found for the compounds (Sp)-3aa-1 or (Rp)-3aa-1.

According to the reviewer's suggestion, several 31P NMR data were remeasured and a clean and sharp 31P spectra were obtained (**1a-2**, **1c**, **1d**, **1f**, **3af**). These 31P spectra have been replaced in the revised Supplementary Information. While 13C NMR was measured again (**1c**, **1d** and **3af**), clearer spectra were not obtained and it was difficult to determine coupling constant (Jc-p) values.

Reviewer 2

While it is regrettable that Reviewer 2 did not offer commendations, we firmly believe that this work represents a groundbreaking molecular transformation with the potential to open new avenues in oligonucleotides therapeutics. Below, we have provided a point-by-point response to the concerns raised by Reviewer 2.

The manuscript by Ohmiya and co-workers describes the development of a synthetic method to access dinucleotides with a tertiary alkylphosphonate internucleotide linkage. The method relies on a light-driven formation of alkylsulfonium derivative, which undergoes an in situ Arbuzov-type reaction with dinucleoside phosphite triester. In addition to the optimization of the reaction conditions and the exploration of its scope and limitations, the Authors demonstrate the feasibility of incorporation of the dinucleoside units with tertiary alkylphosphonate linkage into oligonucleotides. Thus, the paper needs to be evaluated at two levels: (1) the novelty of the developed synthetic method and (2) the importance and potential implications of the new modification of oligonucleotides.

From the chemistry viewpoint, the Authors employ their resourceful in-house approach of generating tertiary carbocation-like sulfonium species via the light induced activation of redox-active esters. It has been extensively explored earlier and shown to efficiently deliver tertiary alkyl groups to a variety of nucleophiles (not phosphorus-based, though). From this perspective, the presented results constitute only an incremental advancement and moderate originality, as basically

yet another nucleophile family has been tested. Nevertheless, the method provides access to tertiary alkylphosphonates, which cannot be easily obtained otherwise. The only general method existing up to now toward this class of products is phospho-Michael addition, which is limited to compounds with electron-poor tertiary alkyl moiety, while the reported reaction allows for the incorporation of electron-rich groups.

In this present study, we focused on development of synthetic protocol for tertiary alkylation on phosphorus atoms of dinucleotides. Therefore, we relied on our solid carbocation-like sulfonium species. Although we tested other carbocation species generated by other light-driven RPC catalysis or Lewis acid catalysis (Fig. 3B), the desired products were not obtained at all. Under this review process, Aggarwal and co-workers introduced phosphorus radical trap that can produce alkyl phosphonates through the reaction with alkyl radicals, but tertiary alkylation was unsuccessful (*J. Am. Chem. Soc.* **2023**, *145*, 18649–18657). Therefore, our protocol based on tertiary carbocation-like sulfonium species stands as the sole one performing tertiary alkylation on phosphorus atoms of dinucleotides. We believe that the outstanding reactivity demonstrated in this manuscript must have important implications on the chemical modification of various biomolecules as well as dinucleotides.

The Authors made a choice to optimize the reaction conditions directly for a dinucleotide phosphite substrate. This challenging starting material necessitated the use of a stoichiometric amount of benzo[*b*]phenothiazine reagent. Also, the obtained yields are mostly moderate.

Although this protocol requires a stoichiometric amount of benzo[*b*]phenothiazine reagent, the optimal *N*-methyl benzo[*b*]phenothiazine (**PTH2**) could be synthesized from inexpensive raw materials without expensive transition metals such as palladium.

I do not fully agree with this approach, because the method could be made more general, and perhaps even catalytic for simpler phosphite esters (the corresponding products would also be of interest). In my opinion, this would benefit the chemical community more, rather than focusing on the nucleotide derivatives alone.

Thank you for the valuable comments. As pointed out by the reviewer, tertiary alkylation of simple phosphite esters proceeded with the catalytic amount of photoredox catalyst to afford the desired products in high yields. (The representative example has been added as

Supplementary Fig. 7A in the revised Supplementary Information). This result implies the further application of our protocol for synthesis of alkyl phosphonates.

Supplementary Fig. 7. Examination of Catalytic Reactions

Finally, a bit aside, I would like to ask if the Authors have tried or considered employing H-phosphonate diesters instead of phosphite triesters as the nucleophiles. The former have several advantages, such as straightforward synthesis (also in the case of dinucleotides) and rendering the auxiliary beta-cyanoalkyl substituent unnecessary.

Thank you for the valuable comment. We examined the reaction with use of simple H-phosphonate diester as a P nucleophile, but unfortunately the reaction did not proceed. The corresponding data has been added as Supplementary Fig. 7B in the revised Supplementary Information.

Supplementary Fig. 7. Examination of Catalytic Reactions

Coming to the second aspect, the development of oligonucleotide therapeutics requires utilizing modified nucleotide units to attain desired pharmacological characteristics. This is of particular importance for the antisense and aptamer technologies, wherein, unlike in the case of mRNA vaccines, the enzymatic stability of the oligonucleotide needs to be considerably augmented. In this context,

expansion of the toolbox of available modifications, especially these at the internucleotide phosphate linkage, is a viable direction of research. The tertiary alkylphosphonate moiety introduced in the work seems to bring about many favorable properties, including good hydrolytic stability and charge-neutrality. Very importantly from the practical perspective, the modified dinucleotide units can be efficiently incorporated into oligonucleotides via the standard phosphoramidite solid-phase synthesis (although a prior swap of protecting groups is necessary). Also, the high hydrolytic stability (and that toward F-) of the modification is nicely demonstrated. A clear highlight is the possibility to separate the P-diastereomers and use them individually for the modification of specific positions of the oligonucleotide. However, the actual usefulness of tert-Alk-P-modified oligos for biochemical and pharmaceutical applications is yet to be evaluated. One obvious foreseeable problem is that with this technology only some of the internucleotide linkages can be modified, while at least half of them will remain the regular phosphodiester bonds. In general, there has been a multitude of nucleotide modifications proposed and introduced, but only a handful made it to the therapeutic level.

1. "However, the actual usefulness of tert-Alk-P-modified oligos for biochemical and pharmaceutical applications is yet to be evaluated."

As the reviewers have also acknowledged, the enhancement of chemical stability has been adequately demonstrated for each step of the molecular transformation leading to the phosphoramidite formation and the subsequent oligo synthesis. Furthermore, as evidenced by the results of UPLC retention time analysis, it is apparent that the lipophilicity, the affinity with a C18 column, has significantly increased when compared to other representative neutral backbones (MOP and POEt triester). We believe this highlights the significant uniqueness of our chemical modification and foreseeable usefulness as oligonucleotide therapeutics. The impact of these chemical structures on the properties of oligonucleotides will be reported in subsequent publications.

2. "One obvious foreseeable problem is that with this technology only some of the internucleotide linkages can be modified, while at least half of them will remain the regular phosphodiester bonds."

In this present study, our primary focus was to demonstrate the robustness of the chemistries into oligomers under the representative synthetic conditions. Therefore, the

method exploiting the dimer amidites represents just one approach to synthesize oligonucleotides with the tert-alkyl backbones, and it is potentially expandable to continuous incorporation using techniques such as liquid-phase synthesis. Furthermore, it's worth noting that in the context of nucleic acid therapeutics, there are numerous instances where even a limited number of incorporations can effectively alter pharmacological profiles, as well as pharmacodynamics and toxicity, etc. Therefore, we want to emphasize the potential impact of even a small number of modifications achieved by the methodology demonstrated here in this regard.

3. "In general, there has been a multitude of nucleotide modifications proposed and introduced, but only a handful made it to the therapeutic level."

Thank you for important insight. Regarding the backbone chemistry in oligonucleotides therapeutics, Wave Life Sciences is currently advancing the development of the PN backbone as a novel chemical modification succeeding PS. On the other hand, this represents only the second instance of a backbone chemistry reaching a clinical trial. This highlights the challenges posed by synthetic difficulties and chemical stability, which have historically hindered the application of backbone modifications in drug discovery. We hold the belief that the advancement of foundational chemistry like the reaction developed here will ultimately enhance the success rate in drug discovery.

1) Abstract, line 7: "oligomerization" is not specific enough and should be replaced with "oligonucleotide synthesis". Also, these are not "methyl groups or primary/secondary alkyls" that degrade, but rather the phosphonate moieties.

According to the reviewer's advice, the sentences have been modified in the revised manuscript.

phosphodiester backbones in these compounds. However, at present, the alkyl moieties that can be attached to phosphorus atoms in these compounds are limited to methyl groups or primary/secondary alkyls, and such alkylphosphonate moieties can degrade during oligonucleotide synthesis. The present work demonstrates the *tertiary* alkylation of the phosphorus

2) Page 2, line 8: One cannot say that oligonucleotides are composed of "dinucleotide subunits", as there is nothing distinct about 2-nucleotide units (why not "trinucleotide subunits"?). Change to just "nucleotide" subunits.

According to the reviewer's advice, we changed the phrase to "nucleotide subunits" instead of "dinucleotide subunits".

3) Page 5, after the information about the diastereomers: The Authors may add a comment on the probable stereochemical course of the reaction. For instance, that probably each P-stereoisomer of the substrate is converted into the corresponding P-stereoisomer of the products, and as a mixture was used for the reaction, a mixture is obtained.

Thank you for the valuable suggestion. According to the reviewer's advice, the information about the diastereomers have been added to the revised manuscript.

(Table 1, entry 1). ~~This compound was obtained as a mixture of diastereomers that were separated using a silica gel column after which the structure of one of the diastereomers was determined by X-ray analysis. Since phosphite 1a-1 was used as a mixture of diastereomers and each P-stereoisomer of the substrate was converted to the corresponding P-stereoisomer of the product, 3aa was obtained as a mixture of diastereomers. After separation using a silica gel column chromatography, the structure of one of the diastereomers was determined by X-ray analysis.~~ This analysis showed that the phosphorus

4) Table 1: It would improve the readability if a column listing the R1 and R2 substituents was added to the table.

According to the reviewer's suggestion, the R¹ and R² substituents have been specified in Table 1 in the revised manuscript.

Table 1 | Screening of leaving groups^a

Entry	Phosphite substrate	Yield (%) of 3aa ^b
1	1a-1: R ¹ = Ph, R ² = H	71
2	1a-1: R ¹ = Ph, R ² = H	20
3	1a-2: R ¹ , R ² = Me	68
4	1a-3: R ¹ = Me, R ² = H	trace
5	1a-4: R ¹ , R ² = H	trace

5) Figure 2: Please add the isolated yield of 3aa

According to the reviewer's advice, the data of 3aa has been added to Figure 2.

6) Page 10: I would change the heading from “Mechanistic studies” to “Control experiments”. True mechanistic studies would be more elaborate, like proper kinetic measurements, etc.

According to the reviewer’s advice, the heading of Figure 3 has been changed to “Control experiments”.

Reviewer 3:

1. Have the authors tried used an external reversible-trapping reagent for the tertiary cation (instead of stoichiometric amount of photocatalyst)? Maybe screening of the trapping reagent could lead to better results.

Thank you for the valuable suggestion. We performed the reaction using a catalytic amount of a phenothiazine photoredox catalyst and a stoichiometric amount of sulfilimine, used in the stabilized cation pool method (Hayashi, R.; Shimizu, A.; Yoshida, J. *J. Am. Chem. Soc.* **2016**, *138*, 8400–8403), as a trapping reagent for the carbocation, but the yield was not improved. The corresponding data has been added as Supplementary Fig. 8A in the revised Supplementary Information.

Supplementary Fig. 8. Examination of the Addition of Trapping Reagent for Carbocation⁴

2. Have the authors tried using a simpler phosphite, for example with a methoxy group, while adding an external nucleophile to abstract the methyl group from the phosphonium intermediate to enable the Arbuzov-type reactivity?

Thank you for the valuable suggestion. We tried to examine a simpler phosphite with a methoxy group as the leaving group, but it could not be synthesized because it was quickly oxidized.

When simple trihexylphosphite (**1g**) was used as a nucleophile, Arbuzov-type alkylation proceeded with a catalytic amount of benzo[*b*]phenothiazine and lithium tetrafluoroborate to afford **3ga** in high yield. The corresponding data has been newly added as Supplementary Fig. 7A in the revised Supplementary Information.

Supplementary Fig. 7. Examination of Catalytic Reactions⁴

3. The substrate scope shows rather poor compatibility with trialkyl-substituted tertiary carboxylic acid. What are the results with these substrates? As I suggested

above, maybe an extra trapping reagent could lead to the success with these substrates.

Thank you for the valuable suggestion. The reaction did not proceed with trialkyl-substituted tertiary carboxylic acid derivatives (Supplementary Fig. 9). The instability of the corresponding carbocation equivalent was thought to be the problem. The addition of an extra trapping reagent was also examined, but no alkylated product was obtained (Supplementary Fig. 8B).

4. It will be nice if the authors can indicate any unsuccessful substrates, which will provide a clearer picture for the synthetic community.

Thank you for the valuable suggestion. According to the reviewer's advice, we showed the data for the unsuccessful substrates in the revised Supporting Information (Supplementary Fig. 9).

5. Further potential mechanistic study could include distinguishing between outer-sphere single electron transfer mechanism and EDA mechanism. The benzophenothiazine is a pretty good donor and RAE is known to be good acceptor.

Thank you for the valuable suggestion. In our previous paper, we measured the UV/vis absorption spectrum of benzo[*b*]phenothiazine coexisting with RAE (Shibutani, S.; Nagao, K.; Ohmiya, H. *Org. Lett.* **2021**, *23*, 1798–1803). In the present study, a significant red shift in the absorption band was observed, suggesting the involvement of the EDA complex in the SET step.

6. Some very closely related automated synthesis study should be cited, e.g. *Nat. Commun.* 2021, *12*, 4396; *Nat. Chem.* 2021, *13*, 451; *ACS Cent. Sci.* 2022, *8*, 205.

Thank you for the valuable suggestion. According to the reviewer's advice, the corresponding references were added to the revised manuscript as references 42, 43, and 44, respectively.

REVIEWERS' COMMENTS

Reviewer #1 (Remarks to the Author):

I am satisfied with the changes made and believe the manuscript can now be published.

Reviewer #2 (Remarks to the Author):

The Authors more than satisfactorily addressed all the specific points raised by me and the other Referees. The manuscript is now flawless.

In particular, I appreciate performing the experiment with a simple phosphite triester, which worked beautifully. In my opinion, this additional result turns the scale in favor of accepting the manuscript for publication. Namely, it shows that the developed chemistry is general and, thus, it may find synthetic applications outside of the very special area of modified oligonucleotides.

In the light of the above, I recommend to publish the work by Ohmiya and co-workers as is in Nature Communications.

Reviewer #3 (Remarks to the Author):

The authors have addressed all my concerns, regarding mechanistic study, substrates scope and alternative use of reagents, in the respond letter with corresponding experimental results. These newly added results into the supplementary information could provide a better understanding of this novel synthetic method. As of now I have no further questions and reckon this manuscript is ready for publication in Nature Communications.

Thank you for the reviewers' comments. I appreciate valuable comments from the reviewers. The reviewers don't require further revisions.

Sincerely yours,

Hirohisa Ohmiya

Professor

Kyoto University